# Comparative Analysis of Cholinergic Machinery in Carcinomas: Discovery of Membrane-Tethered ChAT as Evidence for Surface-Based ACh Synthesis in Neuroblastoma Cells

**DOI:** 10.3390/ijms262110311

**Published:** 2025-10-23

**Authors:** Banita Thakur, Samar Tarazi, Lada Doležalová, Homira Behbahani, Taher Darreh-Shori

**Affiliations:** Division of Clinical Geriatrics, Department of Neurobiology, Care Sciences and Society, Center for Alzheimer Research, Karolinska Institutet, SE-141 86 Stockholm, Sweden; banita.thakur@ki.se (B.T.); samar.m.tarazi@hotmail.com (S.T.); lada.dolezalova@oist.jp (L.D.); homira.behbahani@ki.se (H.B.)

**Keywords:** cholinergic system, choline acetyltransferase (ChAT), acetylcholine (ACh), cancer, neuroblastoma, lung cancer, flow cytometry, confocal microscopy, neurodegeneration, Alzheimer’s Disease (AD), amyotrophic lateral sclerosis (ALS)

## Abstract

The cholinergic system is one of the most ancient and widespread signaling systems in the body, implicated in a range of pathological conditions—from neurodegenerative disorders to cancer. Given its broad relevance, there is growing interest in characterizing this system across diverse cellular models to enable drug screening, mechanistic studies, and exploration of new therapeutic avenues. In this study, we investigated four cancer cell lines: one of neuroblastoma origin previously used in cholinergic signaling studies (SH-SY5Y), one non-small cell lung adenocarcinoma line (A549), and two small cell lung carcinoma lines (H69 and H82). We assessed the expression and localization of key components of the cholinergic system, along with the cellular capacity for acetylcholine (ACh) synthesis and release. Whole-cell flow cytometry following membrane permeabilization revealed that all cell lines expressed the ACh-synthesizing enzyme choline acetyltransferase (ChAT). HPLC-MS analysis confirmed that ChAT was functionally active, as all cell lines synthesized and released ACh into the conditioned media, suggesting the presence of autocrine and/or paracrine ACh signaling circuits, consistent with previous reports. The cell lines also demonstrated choline uptake, indicative of functional choline and/or organic cation transporters. Additionally, all lines expressed the ACh-degrading enzymes acetylcholinesterase (AChE) and butyrylcholinesterase (BChE), as well as the alfa seven (α7) nicotinic and M1 muscarinic ACh receptor subtypes. Notably, flow cytometry of intact SH-SY5Y cells revealed two novel findings: (1) ChAT was localized to the extracellular membrane, a feature not observed in the lung cancer cell lines, and (2) BChE, rather than AChE, was the predominant membrane-bound ACh-degrading enzyme. These results were corroborated by both whole-cell and surface-confocal microscopy. In conclusion, our findings suggest that a functional cholinergic phenotype is a shared feature of several carcinoma cell lines, potentially serving as a survival checkpoint that could be therapeutically explored. The discovery of extracellular membrane-bound ChAT uniquely in neuroblastoma SH-SY5Y cells points to a novel form of in situ ACh signaling that warrants further investigation.

## 1. Introduction

The cholinergic signaling system, which relies on acetylcholine (ACh), is one of the most widespread and evolutionarily conserved communication systems in the body. Cholinergic cells express the enzyme choline acetyltransferase (ChAT), which synthesizes ACh by transferring the acetyl group from acetyl-coenzyme A (A-CoA) to choline. The canonical view is that ChAT is a cytosolic enzyme, with ACh biosynthesis occurring exclusively in the cytoplasm. However, recent studies challenge this view, demonstrating that ChAT can be released by lymphocytes and astrocytes into extracellular fluids, where it contributes to maintaining extracellular ACh equilibrium even in the presence of active ACh-degrading enzymes [1,2].

In addition to ChAT, the cholinergic system comprises two major receptor families: nicotinic acetylcholine receptors (nAChRs), which are ligand-gated ion channels, and muscarinic acetylcholine receptors (mAChRs), which are G protein-coupled receptors [3]. Two key transporter proteins also support cholinergic function: the vesicular ACh transporter (VAChT), responsible for packaging ACh into synaptic vesicles, and the high-affinity choline transporter (hChT), which facilitates choline reuptake following ACh hydrolysis at synapses. ACh signaling is terminated by enzymatic hydrolysis into choline and acetate [3]. Acetylcholinesterase (AChE) has been considered to be the primary enzyme responsible for ACh hydrolysis while butyrylcholinesterase (BChE) may act as a decoy enzyme, protecting AChE from dietary or environmental inhibitors [4].

Cells that express ChAT and synthesize ACh are classified as cholinergic, while those expressing ACh receptors are termed cholinoceptive. Importantly, cholinergic cells can also be cholinoceptive, as ACh may exert autocrine effects via autoreceptors [3].

Cholinergic cells are broadly categorized into neuronal and non-neuronal types. Neuronal cholinergic systems include the central cholinergic network originating from the basal forebrain, comprising four major nuclei (Ch1–Ch4) that project throughout the brain [3,5]. This system plays a critical role in cognition, attention, and memory, and is notably affected in dementia disorders such as Alzheimer’s disease (AD), dementia with Lewy bodies, and Parkinson’s disease dementia [6,7,8,9,10]. Cholinergic interneurons within the nigrostriatal system, particularly in the putamen, are compromised in Parkinson’s disease and related disorders such as corticobasal degeneration and progressive supranuclear palsy [11,12,13,14]. Another neuronal cholinergic system comprises the parasympathetic system, consisting of the 12 cranial nerves which connect directly to organs, muscles, and glands, controlling various bodily functions. Most of these together with cholinergic neuromotor neurons in the brainstem and spinal cord are progressively lost in neuromotor disorders like amyotrophic lateral sclerosis (ALS) [15,16]. Another major neuronal cholinergic system is the enteric nervous system (ENS), where cholinergic neurons regulate gastrointestinal motility and secretion. It is estimated that over 60% of ENS neurons are cholinergic, and emerging evidence suggests that these circuits are affected in various neurodegenerative diseases [17,18].

Non-neuronal cholinergic cells include a diverse array of ChAT-expressing cell types such as epithelial, mesothelial, endothelial, and immune cells [19]. Notably, ChAT expression increases in lymphocytes and macrophages under inflammatory conditions, forming part of the “cholinergic anti-inflammatory pathway,” in which ACh modulates cytokine release and inflammation [1,19]. Impairment of this pathway in aging and disease may contribute to excessive inflammatory responses. Increasing evidence also points to upregulation of cholinergic machinery in various cancers, particularly small cell lung cancer (SCLC) and colon cancer [20,21,22,23,24]. In these contexts, ACh may function as an autocrine growth factor [23,25], and nicotinic receptor antagonists have been proposed as potential anticancer agents [20,26].

Although ChAT is generally considered a soluble cytosolic enzyme, reports have described intracellular membrane-bound forms of ChAT in synaptosomes and cholinergic nerve terminals across several species [27,28,29]. As an integral membrane protein, ChAT may exhibit distinct biochemical properties compared to its cytosolic counterpart [29]. In a recent in vitro study, we demonstrated that human ChAT readily embeds into micellar nanoparticles, resulting in a dramatic 10-fold increase in its catalytic activity [30]. ChAT also retains a nuclear localization signal, suggesting potential translocation to the nucleus, although its nuclear function remains unclear [31,32].

In this study, we investigated the expression of ChAT and related cholinergic markers in four human tumor cell lines: neuroblastoma (SH-SY5Y), lung adenocarcinoma (A549), and two small cell lung cancer lines (H69 and H82) [33]. Our aim was to characterize and establish robust cholinergic model systems for future screening and in vitro evaluation of ChAT ligands, including inhibitors and activators previously reported by us [30,34,35], with potential applications as anti-tumor agents. Additionally, we sought to determine whether ChAT is localized extracellularly as a membrane-anchored enzyme in these tumor cells, as we previously observed in human spermatozoa [2]. We further examined the presence and localization—surface versus intracellular—of key components of the cholinergic system across the four cancer cell lines. Our findings provide novel insights into cholinergic signaling mechanisms that may be relevant to both neurodegeneration and carcinoma pathology.

## 2. Results

### 2.1. Differential ChAT Localization in a Neuroblastoma Cell Line Compared with a Lung Adenocarcinoma and Two Small Cell Lung Carcinoma Cells Lines

The small cell lung carcinoma (SCLC) cell lines H69 and H82, the lung adenocarcinoma cell line A549, and the neuroblastoma cell line SH-SY5Y were evaluated for ChAT cell surface expression without fixation or permeabilization (Figure 1). Cells were simultaneously incubated with both the anti-ChAT antibody and the Live/Dead dye for surface flow cytometry. This was done to ensure that plasma membrane integrity was not compromised, particularly since two of the cell lines (SH-SY5Y and A549) are adherent and required trypsin treatment to form a suspension. The Live/Dead dye selectively stains cells with compromised membranes, and the resulting dot plots in Figure 1 show negligible signal in quadrants Q1 and Q2, indicating that SH-SY5Y cells retained intact plasma membranes following trypsin treatment. The same was observed for the other three cell lines, including A549, which, like SH-SY5Y, grows as an adherent cell line. Interestingly, under these conditions, surface ChAT localization was exclusively detected in the SH-SY5Y neuroblastoma cell line, but not in the other cell lines. This surface localization was abolished when the antibody was pre-blocked with recombinant ChAT protein, confirming the specificity of the staining (Figure 1).

Next, whole-cell staining was performed using fixation and permeabilization to assess the total ChAT expression within the cells, encompassing both intracellular and extracellular pools of ChAT. Positive intracellular ChAT expression and localization were observed across all four tested cell lines (Figure 2), consistent with previous reports. ChAT staining using the pre-blocked antibody with recombinant ChAT protein significantly reduced the signal across all cell lines, including H69 and A549 (as shown in Appendix A), further confirming the specificity of the antibody for ChAT as the target protein. Notably, since the intracellular staining approach involves cell permeabilization to allow antibody entry, the resulting signal reflects total ChAT expression, as surface membrane-bound ChAT proteins remain accessible to the antibody. Therefore, comparing the proportion of cells positive for surface ChAT staining (66 ± 1%) with those positive for total ChAT staining (79 ± 3%) indicates that a major fraction of ChAT protein in SH-SY5Y neuroblastoma cells is localized extracellularly (compare dot plots in Figure 1 vs. Figure 2). In contrast, in the lung carcinoma cell lines H69, H82, and A549, ChAT staining was only observed following permeabilization, suggesting that ChAT is primarily localized intracellularly in these cells (compare Figure 1 vs. Figure 2). Notably, the relative fluorescence intensity (RFI) of the anti-ChAT antibody compared to the blocked antibody in H82, H69, and A549 is shown in Appendix A. The results indicate that antibody blocking was reasonably effective in H69 and A549 cells, although not as efficient as in H82 or SH-SY5Y cells.

### 2.2. ChAT Is Functionally Intact in Both Neuroblastoma and Lung Cancer Cells

Measuring ACh and choline in an experiment is crucial for assessing the functionality of the cellular cholinergic machinery, as these two molecules are intimately linked within a functional signaling circuit. Their concurrent quantification provides an important measure of the activity and homeostatic state of the cholinergic system, including its potential auto- and paracrine regulatory mechanisms. In particular, it can reveal whether ChAT protein is functionally intact and whether the cells are capable of synthesizing and releasing ACh. We therefore measured the biosynthesis and release levels of ACh in the four cancer cell lines using HPLC-MS/MS analyses. The analysis was performed on both cell lysates and conditioned media from cultured cells, the latter to assess ACh release. Cell-free complete medium (RPMI), with or without FBS, served as negative controls. To preserve biosynthesized ACh during sample preparation, neostigmine—an inhibitor of both AChE and BChE—was used in parallel. The HPLC-MS/MS results, presented in the upper panel of Figure 3, confirmed that all cell lines were capable of synthesizing ACh, as detected in the cell lysates. To normalize and compare ACh and choline levels across cell lines, total protein concentrations in the samples were used. The protein concentration in the cell lysates was 0.58 ± 0.2 mg/mL and did not differ significantly among the cell lines (SH-SY5Y = 0.56 ± 0.10; H82 = 0.52 ± 0.16; H69 = 0.59 ± 0.20; A549 = 0.67 ± 0.28 mg/mL), confirming that the data reflect comparable cell numbers.

Analysis of the conditioned media indicated that SH-SY5Y neuroblastoma cells and the SCLC cell line H82 released substantial amounts of ACh into the medium compared to controls (Figure 3, upper panel, orange bars). In contrast, only low levels of ACh were detected in the conditioned media from H69 and A549 cells (Figure 3, upper panel, orange bars). In summary, all tested cancer cell lines were capable of storing and releasing synthesized ACh.

We further measured choline levels in SH-SY5Y cells and the three lung cancer cell lines, as well as in their conditioned media, using HPLC-MS/MS (Figure 3, lower panel). The concentration of choline in unused culture medium was determined to be 1.8 ± 0.3 μM. Choline levels were generally much higher in the conditioned media than in the cell lysates, indicating that the culture medium was the primary source of choline and that the cells were capable of choline uptake, likely for ACh biosynthesis (Figure 3, lower panel). This also suggests that all tested cell lines express either a specific choline transporter or a general organic cation transporter, or both [36,37].

The presence or absence of neostigmine had no significant effect on choline levels. This likely reflects the fact that choline is not metabolized by either AChE or BChE. Notably, the release and hydrolysis of ACh into choline is unlikely to have significantly influenced the observed choline levels, given that the measured concentrations of ACh were in the nanomolar range (Figure 3, upper panel), which is negligible compared to the micromolar levels of choline detected (Figure 3, lower panel).

### 2.3. Surface Localization Analysis of ChAT and the Related Cholinergic Markers in Neuroblastoma Cells

In addition to ChAT, we performed flow cytometric analyses to assess the expression and localization of selected components of the cholinergic machinery, namely the ACh-hydrolyzing enzymes AChE and BChE, as well as the α7 nicotinic AChR (α7 AChR) and the M1-subtype of muscarinic AChR (M1 AChR) in SH-SY5Y neuroblastoma cells. The results are shown in Figure 4.

Surface flow cytometric analyses indicated that BChE, α7 AChR, and M1 AChR were present at the cell surface of SH-SY5Y neuroblastoma cells, while only minimal levels of AChE were detected extracellularly or at the cell surface (Figure 4; see also Figure 6). These findings partially deviate from prevailing assumptions in the field. For instance, the low surface/extracellular staining of AChE, in contrast to the high BChE staining observed in neuroblastoma cells, challenges the canonical view that AChE is the primary membrane-anchored extracellular enzyme responsible for ACh hydrolysis within synapses [38,39]. Additionally, BChE is generally regarded as a secreted, soluble enzyme, whereas the current surface flow cytometric data clearly demonstrate that BChE is a major surface protein in all four cell lines (Figure 4). Furthermore, all cell lines exhibited positive surface staining for α7 and M1 AChRs, supporting the notion that ACh exerts autocrine activity in these cells [38,39].

### 2.4. Intracellular Expression Analysis of ChAT and Related Cholinergic Markers in Neuroblastoma and Lung Cancer Cells

We next performed flow cytometric analyses following membrane permeabilization of the four cell lines. These analyses are expected to detect the total expression of cholinergic markers, encompassing both surface and intracellular protein pools. The results are presented in Figure 5, which demonstrate that SH-SY5Y neuroblastoma cells were clearly positive for all assessed cholinergic markers (Figure 5A). Essentially, the same pattern was observed in the lung cancer cell lines H69, H82, and A549 (Figure 5B). Notably, fluorescence intensities increased significantly in all cell lines following membrane permeabilization compared to surface staining alone. This is most likely due to the permeabilization step allowing antibodies access to all available binding targets, including those in intracellular compartments. Furthermore, a comparison of fluorescence intensity profiles for AChE and BChE revealed a markedly higher expression of BChE relative to AChE (see the corresponding fluorescence intensity shifts in the histogram panels in Figure 5), corroborating the observations from the flow cytometric surface expression analyses of these enzymes (Figure 4). However, as antibody affinities may differ, this finding should be interpreted with caution.

Similarly, the clear increase in intracellular fluorescence signals for α7- and M1 AChRs compared to non-permeabilized neuroblastoma cells supports the existence of an intracellular pool of these receptors, likely reflecting ongoing synthesis, intracellular trafficking, or receptor internalization processes.

To validate the aforenoted differences in extracellular versus intracellular levels of cholinergic markers in SH-SY5Y neuroblastoma cells, we quantified protein expression using relative fluorescence intensity (RFI), following a previously established protocol [40], as detailed in Section 4.3 of the Materials and Methods. The RFI data (Figure 6A,B) confirmed that ChAT is predominantly localized at the cell surface, indicating its extracellular anchoring, while AChE exhibited minimal expression in both extracellular and intracellular compartments. In contrast, BChE showed notable extracellular presence, although its major fraction was intracellular. Similarly, both α7-nAChR and M1 mAChR were detected at the cell surface in appreciable amounts, yet their predominant localization was intracellular. Importantly, with the exception of ChAT, the surface-localized fraction of each cholinergic protein represented only a minor proportion of its total cellular expression, with whole-cell levels exceeding surface expression by factors of 1.12 (ChAT), 2.56 (AChE), 45.85 (BChE), 36.70 (α7-nAChR), and 84.60 (M1 mAChR), respectively (Figure 6B).

### 2.5. Confocal Microscopy Confirms Extracellular Expression of ChAT Cholinergic in Neuroblastoma

The canonical view is that ChAT is a cytosolic enzyme responsible for synthesizing ACh. However, we recently reported extracellular localization of ChAT at the surface of human sperm using flow cytometry [2]. Interestingly, among the four examined cell lines, only the SH-SY5Y neuroblastoma cells exhibited surface ChAT expression (Figure 1).

To further investigate the observed phenomenon, we performed cell surface confocal microscopy targeting ChAT in the same SH-SY5Y neuroblastoma cell line. Cells were stained under non-permeabilizing conditions to selectively visualize surface-expressed antigens, thereby excluding intracellular signals. For this analysis, three distinct anti-ChAT antibodies—different from the one employed in the flow cytometric assay—were independently utilized to validate the presence of surface-localized ChAT. Confocal imaging of the non-permeabilized cells consistently revealed distinct surface-associated ChAT immunoreactivity with all three antibodies. These findings corroborate the flow cytometric data, providing convergent evidence for the extracellular localization of ChAT in SH-SY5Y cells (Figure 7, Appendix A). Importantly, the signals observed in these confocal analyses are expected to originate exclusively from surface-localized target proteins, as the cells were not permeabilized during staining. As previously noted, Live/Dead staining confirmed that trypsin treatment did not compromise membrane integrity, given that the dye selectively penetrates cells with damaged membranes. As shown in Figure 1, negligible fluorescence signals were detected in quadrants Q1 and Q2, indicating that SH-SY5Y cells retained intact plasma membranes following enzymatic treatment. Nonetheless, to further ensure membrane integrity and eliminate any residual effects of trypsinization, cells were re-seeded directly onto microscopy slides and incubated for 24 h prior to staining, allowing full reattachment and potential membrane repair. This precautionary step ensured that antibody binding was specific to surface antigens and not confounded by unintended membrane disruption.

Similarly, surface confocal microscopy analysis validated the presence of AChE, BChE, M1-mAChR, and α7-nAChR on the surface of SH-SY5Y neuroblastoma cells (Figure 7). These results indicate that SH-SY5Y cells express a functional cholinergic system at the cell surface, suggesting the presence of autocrine ACh activity. Notably, the data suggest that BChE, rather than AChE, may be the predominant surface-anchored cholinesterase (Figure 7, Appendix A, respectively). This observation is in full agreement with the flow cytometry results (Figure 4 and Figure 6).

We next performed whole-cell confocal analyses on permeabilized cells. Confocal microscopy images are presented in Figure 8. For this analysis, as noted before, three distinct anti-ChAT antibodies—different from the one employed in the flow cytometric assay—were independently utilized to validate the presence ChAT (Appendix A). These analyses fully reaffirmed the findings from the whole-cell flow cytometric analysis regarding the expression and cellular distribution of ChAT throughout the cells, including in the cytosol. To ensure antibody specificity, control experiments were conducted using secondary antibodies alone, which yielded no detectable signal (Appendix A), thereby confirming the specificity of the observed staining.

Similar findings were observed for the M1 and α7 acetylcholine receptors (Figure 8) as well as for the acetylcholine-hydrolyzing enzymes BChE (Figure 8, Appendix A) and AChE (Figure 8, Appendix A). Notably, signal intensities for all antibodies were markedly enhanced following cell permeabilization compared to non-permeabilized conditions, indicating substantial intracellular localization, in full agreement with the flow cytometric analysis (Figure 6). This was particularly evident for AChE, which displayed a diffuse intracellular staining pattern post-permeabilization, in contrast to the punctate, dot-like surface expression seen in non-permeabilized cells (compare AChE micrographs in Figure 7 and Figure 8, as well as Appendix A). These results suggest that AChE expression in these cells is predominantly intracellular, with minimal surface localization. Importantly, these observations are fully consistent with the flow cytometric data for AChE presented in Figure 4, Figure 5A and Figure 6.

## 3. Discussion

To the best of our knowledge, this study provides the first compelling evidence for the presence of an extracellularly membrane-bound isoform of ChAT in the SH-SY5Y neuroblastoma cell line. Such an isoform was not observed in the three lung carcinoma cell lines examined. This was carefully confirmed through confocal microscopy analyses of surface-localized ChAT protein. Nonetheless, we have previously reported a similar extracellularly membrane-bound ChAT in human spermatozoa [2].

There are reports indicating the presence of intracellularly membrane-bound ChAT in synaptic vesicles and synaptosomes across several animal species [27,28,29]. Intriguingly, ACh synthesized by membrane-bound ChAT appears to be approximately 40% more resistant to hydrolysis by AChE than ACh produced by the soluble form of ChAT [27]. In a recent study, we also demonstrated that human ChAT has a strong propensity to associate with membrane-like micelles, a phenomenon that enhances its enzymatic activity by more than tenfold [30]. Therefore, the distinct extracellular localization of ChAT in the SH-SY5Y neuroblastoma cell line may represent a novel physiologically relevant cholinergic-enhancing mechanism, potentially involving rapid in situ synaptic recycling of ACh and prolonging its action at the synapse. This interpretation aligns with our previous hypothesis regarding extracellularly membrane-bound ChAT in human spermatozoa, where in situ ACh synthesis was proposed to occur in close proximity to membrane-bound receptors, enabling timely chemotactic signaling to facilitate or regulate sperm motility [2].

For comparison, we also examined ChAT expression in two SCLC neuroendocrine cell lines (H82 and H69) and one NSCLC lung adenocarcinoma cell line (A549). Although these cells lacked detectable extracellularly membrane-bound ChAT, they exhibited intracellular ChAT staining comparable to that observed in SH-SY5Y cells. While only four cancer cell lines were analyzed, our findings suggest that extracellular ChAT localization is not a general feature of cancer cells. To assess whether ChAT expression in these cell lines has functional relevance, we measured ACh production and release. Despite differences in synthesis and release levels among the cell lines, all four were capable of synthesizing and releasing ACh. These findings are consistent with previous reports on ACh and choline production in human colon cancer cell cultures [22], as well as in SCLC, where the H82 cell line—similar to our study—exhibited higher ACh levels than H69 cells [20].

The cholinergic machinery includes, among other components, various forms of nicotinic and muscarinic ACh receptors (nAChRs and mAChRs), as well as ACh-degrading enzymes such as AChE and BChE. Many of these components have previously been reported in SCLCs [20,25,41] and in the neuroblastoma SH-SY5Y cell line [24,42]. We therefore examined several of these components in our study. In SH-SY5Y cells, we first analyzed the presence of surface-localized AChE, BChE, M1-mAChR, and α7-nAChR using both flow cytometry and confocal microscopy. Both M1-mAChR and α7-nAChR were detected extracellularly, which is expected given their known integration into the plasma membrane and previous reports of their presence in various cancer cells [23,25,42]. The observation of extracellularly membrane-bound BChE, rather than AChE, is particularly noteworthy. If confirmed by further experiments, this finding would challenge the prevailing view that AChE is the primary membrane-anchored ACh-hydrolyzing enzyme found extracellularly in multimeric forms—such as on blood cells, at neuromuscular junctions, and within synaptic clefts of cholinergic interfaces in the brain [38,39]. BChE, by contrast, has traditionally been considered a soluble enzyme. However, recent studies have suggested the existence of membrane-anchored AChE-BChE hybrid molecular forms [43]. While our findings support this notion, the negligible or absent surface staining of AChE in SH-SY5Y cells suggests that membrane-anchored BChE is the dominant form in this cell line.

Upon membrane permeabilization, flow cytometric analyses reflect antibody interaction with both surface and intracellular protein pools. As shown in Figure 6B, a comparison of relative fluorescence intensity between total and surface AChE and BChE signals in SH-SY5Y cells suggested that BChE was significantly more abundant than AChE, supporting the conclusion that BChE is the predominant membrane-anchored cholinesterase. Confocal microscopy further corroborated this (Figure 7 and Appendix A), showing an even distribution of BChE across the cell surface, whereas AChE displayed a punctate, localized pattern suggestive of discrete focal expression. Nonetheless, this conclusion should be interpreted with caution, as different antibodies targeting distinct proteins were used, and variations in antibody affinity may influence staining intensity.

Furthermore, we investigated the expression of cholinergic signaling biomolecules in lung cancer cell lines, specifically the SCLCs H82 and H69, as well as the lung adenocarcinoma cell line A549. The results closely resembled those observed in the neuroblastoma SH-SY5Y cell line, not only in terms of ChAT expression and ACh synthesis and release, but also regarding the expression patterns of AChE, BChE, M1-mAChR, and α7-nAChR. Notably, in all three cell lines, the total staining intensity of AChE appeared significantly lower than that of BChE (Figure 6B, also see the Appendix A). Interestingly, previous reports have described hybrid BChE-AChE molecular forms in human glioma [44].

Given that the examined cell lines are of cancerous origin, this phenomenon may represent a distinct feature of cancer pathology. For example, BChE is known to possess scavenging properties against cytotoxic agents naturally ingested through sources such as food. Thus, elevated BChE expression may provide tumor cells with a protective mechanism against such agents. Indeed, one study reported increased BChE levels in the serum of patients with cancers of various tissue origins following treatment, compared to baseline levels or healthy controls [45].Our findings suggest that screening current cytotoxic agents for their susceptibility to BChE metabolism is warranted, to identify compounds that are not metabolized by BChE and may therefore be more effective under these conditions. Moreover, the presence of key components of the cholinergic system—including its central enzyme, ChAT—raises the possibility of a significant role for cholinergic signaling in cancer cell biology. However, based on our current data, the presence or expression of cholinergic components should not be interpreted as causative of cancer, as no normal human cell lines were included for comparison. Nevertheless, our findings align with existing reports indicating that cholinergic signaling promotes cell survival and proliferation in various cancers, including lung cancer [23,24,26,36,37,42,46,47]. Therefore, further research into the role of cholinergic signaling in cancer biology is imperative, as it may contribute to the development of tolerance to therapeutic insults such as chemo- and/or radiotherapy. Inhibiting this pathway could potentially enhance the efficacy of such treatments.

We have previously hypothesized that ACh may also function as an intracellular signaling molecule involved in monitoring mitochondrial bioenergetics and regulating mitochondria-driven apoptosis [48], a process that may be differentially altered in cancer and neurodegenerative diseases such as AD and ALS. According to our hypothesis, intracellular ACh plays a critical role in protecting cells from mitochondria-driven apoptosis (Figure 9). This could help explain how two seemingly distinct disorders may share a common pathological feature. In neurodegenerative diseases (e.g., AD and ALS), selective degeneration occurs in central cholinergic neurons, spinal and peripheral cholinergic motor neurons, and parasympathetic cranial nerves, accompanied by a marked reduction in ChAT expression [6,7,8,9,10,11,12,13,14,15,16]. Consequently, the expected decline in intracellular ACh biosynthesis via ChAT may reduce cellular protection against mitochondrial apoptosis, thereby facilitating neuronal degeneration. In contrast, tumor cells appear to upregulate a pronounced cholinergic phenotype to shield themselves from apoptosis [20,25,41].

Maintaining continuous intracellular ACh biosynthesis for such protective purposes is a highly energy-demanding process, as it requires equimolar amounts of acetyl-CoA—normally used for ATP production. This suggests that upregulated ACh signaling in tumor cells may represent a strategic energy investment, especially considering the high energy demands of cancer cell proliferation. Thus, the acetylcholine–mitochondria–apoptosis axis emerges as a critical area of investigation for understanding both selective cholinergic neurodegeneration in dementia and neuromotor disorders, and the survival strategies employed by tumor cells, including neuroblastoma and lung cancer.

An important question that remains unresolved in the current study is whether ChAT is fully localized extracellularly—similar to how AChE is anchored to the membrane via specific anchoring proteins [38,39],—or whether it is partially embedded within the plasma membrane, as previously demonstrated in micellar particles [30], synaptic vesicles, and synaptosomes [27,28,29]. In our flow cytometric analysis, we used an anti-ChAT antibody (Abcam Cat# ab181023, RRID: AB_2687983) that recognizes an epitope near the C-terminal domain of the ChAT protein (around amino acid 700). This suggests that the C-terminal region of ChAT is exposed on the extracellular surface of the plasma membrane. To further investigate ChAT localization, we employed three additional antibodies in confocal analyses. One was a monoclonal mouse anti-ChAT antibody (MAB3447, R&D Systems), which targets a region between Ala2 and Pro630 (Accession # NP_066266). However, this epitope range does not allow us to determine whether the antibody binds to the N-terminus, C-terminus, or an internal domain of the protein. The other two antibodies—rabbit polyclonal anti-ChAT antibodies from Abnova (PAB14536) and Millipore (AB143)—likely contain subclones that recognize multiple epitopes across the ChAT sequence, further complicating precise localization. Nonetheless, our findings suggest that ChAT is either embedded in the plasma membrane or fully extracellular but anchored to the membrane, as it was not washed away during surface staining procedures. This contrasts with our previous observations of sperm surface ChAT, which was readily removed under similar conditions [2]. The retention of ChAT on the cell surface implies a stable membrane association, but the nature of this interaction—whether via transmembrane domains, lipid anchors, or protein–protein interactions—remains to be elucidated. Taken together, while our data strongly support the presence of extracellularly exposed ChAT, the exact membrane topology and anchoring mechanism remain uncertain. Addressing this question will require additional studies, including epitope mapping, protease protection assays, and advanced imaging techniques such as cryo-electron microscopy or proximity labeling. Clarifying the localization and membrane association of ChAT is essential for understanding the mechanism of in situ ACh signaling and its potential role in cancer cell survival, communication, and therapeutic targeting.

In summary, this study provides compelling evidence for the presence of an extracellular, membrane-bound form of ChAT exclusively in neuroblastoma cells, in contrast to the three lung cancer cell lines tested. However, all examined cell lines expressed intracellular ChAT. Additionally, all cells expressed several other components of the cholinergic machinery, reinforcing the notion that ChAT expression serves a biological function consistent with ACh signaling. These findings highlight the complexity of cholinergic signaling, particularly the distinct roles of membrane-bound and intracellular cholinergic markers. Dysregulation of these components has been linked to neurodegenerative diseases such as Alzheimer’s disease, Parkinson’s disease, and amyotrophic lateral sclerosis. Membrane-bound and intracellular cholinergic markers influence ACh homeostasis, receptor trafficking, apoptotic pathways, and neuroinflammation, making them critical targets for therapeutic intervention. Further research is needed to elucidate the molecular pathways regulating these markers and to assess their potential as therapeutic targets in both neurodegenerative diseases and cancer. A deeper understanding of their interactions and regulatory mechanisms could pave the way for strategies aimed at restoring cholinergic system integrity and mitigating disease progression.

## 4. Materials and Methods

### 4.1. Reagents and Antibodies

This study employed a range of reagents and antibodies essential for cell culture and flow cytometric analysis. RPMI-1640 medium, fetal bovine serum (FBS), trypsin, penicillin, and streptomycin were sourced from Thermo Fisher Scientific (Stockholm, Sweden), along with fixation and permeabilization buffers (Cat# 00-8333-56). Cell culture containers, including T25 and T75 flasks, plates, and 15 mL Falcon tubes, were obtained from Corning (via Merck, Darmstadt, Germany).

For the detection of ChAT via flow cytometry, an APC-conjugated recombinant anti-ChAT antibody (Cat# AB224001, Abcam (Cambridge, UK)) was used, which targets a sequence at amino acid 700 [https://www.antibodyregistry.org/AB_2687983 (accessed on 16 October 2025)]. Additional cholinergic markers were analyzed using FITC-conjugated anti-AChE antibody (A-11, Sc-373901), PE-conjugated anti-BChE antibody (D-5, Sc-377403), AF647-conjugated anti-α7 AChR antibody (319, Sc-58607), and PE-conjugated anti M1 AChR antibody (G-9, Sc-365966), all from Santa Cruz Biotechnology (Dallas, TX, USA).

Surface staining was performed without fixation, using a 1:1500 dilution of LIVE/DEAD™ Cell Stain dye (Cat# L23101, Invitrogen, Thermo Fisher Scientific (Stockholm, Sweden)) to assess membrane integrity, as previously described [2]. For intracellular staining, cells were treated with permeabilization buffer (Cat# 88-8824-00, eBioscience™, via Thermo Fisher Scientific) following the manufacturer’s protocol. Recombinant human ChAT protein (5.1 mg/mL) was produced and purified by the Protein Science Facility at Karolinska Institutet (Stockholm, Sweden), as previously reported [34].

### 4.2. Cell Lines

The study utilized four human cell lines: two small cell lung carcinoma lines (NCI-H69 and NCI-H82), one lung adenocarcinoma line (A549), and one neuroblastoma line (SH-SY5Y), all obtained from the American Type Culture Collection (ATCC, Manassas, VA, USA). Cells were cultured in RPMI-1640 medium supplemented with 10% FBS and 1% penicillin-streptomycin, and maintained at 37 °C in a humidified incubator with 5% CO_2_. Adherent cell lines (A549 and SH-SY5Y) were detached using 0.25% trypsin and 0.02% EDTA once they reached 80–90% confluency. In contrast, the non-adherent SCLC lines (H69 and H82) were maintained in suspension and processed by centrifugation and washing without enzymatic detachment.

Cell Line Selection Rationale: We selected the SH-SY5Y neuroblastoma cell line due to its extensive characterization in the literature and the availability of well-established differentiation protocols [42], which have been successfully implemented in our laboratory. This cell line will be utilized in upstream studies investigating ChAT ligands as potential therapeutics for neurodegenerative disorders such as AD and ALS. Additionally, the small-cell lung cancer cell line (H69 and H82) was chosen because disease relapses following an initially favorable response to chemotherapy remains a significant clinical challenge. Despite ongoing research, few novel therapeutic targets have demonstrated meaningful clinical efficacy [49], underscoring the urgent need for innovative approaches.

### 4.3. Assessment of Cholinergic Markers by Flow Cytometry

Flow cytometric analysis was conducted to assess the expression of cholinergic markers in both adherent and suspension cell lines. For A549 and SH-SY5Y cells, staining was performed after detachment, washing in PBS, and cell counting. Suspension cells (H69 and H82) were directly centrifuged, washed, and counted. Approximately 1 × 10^6^ cells were used for each staining condition.

Surface staining involved simultaneous incubation of cells with LIVE/DEAD™ dye and the respective conjugated antibody in PBS containing 0.5% BSA, for 30 min at 4 °C. These cells were not fixed or permeabilized. Whole-cell staining was performed following fixation in 4% paraformaldehyde for 10 min at room temperature, washing, and permeabilization according to the manufacturer’s instructions. Cells were then incubated with primary antibodies at specified dilutions for 30 min at 4 °C, washed, and resuspended in PBS for analysis using a CytoFLEX S flow cytometer (Beckman Coulter, Stockholm, Sweden).

To validate the specificity of ChAT antibody staining, cells were incubated with recombinant ChAT protein (~102 μg/mL) alongside the anti-ChAT antibody (1:1250 dilution) before staining. Similar protocols were applied for AChE, BChE, α7-AChR, and M1-AChR, using a 1:50 dilution for each antibody.

All experiments were performed in triplicate. In selected experiments, protein expression was quantified as relative fluorescence intensity (RFI), calculated by dividing the geometric mean fluorescence intensity (GM-FI) of the marker by that of its control, using FlowJo software (v10.10.0). For ChAT, the control consisted of cells stained with the blocked antibody, while for other markers, cells stained only with LIVE/DEAD dye served as controls. To compare total versus surface expression, RFI was calculated by dividing GM-FI from whole-cell staining by GM-FI from surface staining of the same marker.

### 4.4. Assessment of ACh Levels by High-Performance Liquid Chromatography (HPLC)

To assess acetylcholine (ACh) levels, cells were cultured in 12-well plates for 24 to 48 h. For non-adherent cell lines (H69 and H82), the entire contents of the wells were transferred to Falcon tubes and centrifuged at 3000 rpm. From these, 1–2 mL of the supernatant—representing the conditioned medium—was collected for subsequent ACh and choline measurements. In the case of adherent cell lines (SH-SY5Y and A549), the conditioned medium was collected directly using a pipette and transferred to Eppendorf tubes. The cells were then detached via trypsinization, pelleted by centrifugation, and washed with PBS. After cell counting, the pellets were homogenized using a steel bead homogenizer with a 25 s ON/5 s OFF cycle repeated twice, yielding cell lysates.

These lysates were centrifuged at 15,000 rpm for 10 min at 4 °C, and the resulting supernatants were transferred to fresh Eppendorf tubes. To precipitate proteins, 100 µL of pre-chilled acetonitrile (stored at −20 °C) was added to both the cell lysates and conditioned media, followed by vortexing until the solutions turned cloudy. Samples were then centrifuged at 17,000× *g* for 20 min at 4 °C. The clarified supernatants were stored on dry ice until HPLC-MS/MS analysis. For quantification, 50 µL of each sample was transferred to injection vials, and 10 µL of a tuning solution containing 1 µg/mL acetylcholine-d9 chloride was added to each vial. As controls, samples of complete RPMI with FBS and RPMI without FBS (both without cells) were collected and processed in parallel.

### 4.5. Immunofluorescence Staining of SH-SY5Y Cells for ChAT and Related Cholinergic Markers

Immunofluorescence staining was performed on SH-SY5Y cells to visualize ChAT and other cholinergic markers. Once cells reached near confluency, they were trypsinized for five minutes and seeded at a density of 50,000–70,000 cells per 35 mm dish onto 20 mm glass-bottom wells (P35G-1.5-20-C, MatTek Corporation, Ashland, MA, USA). Cells were incubated for 24 h to ensure intact plasma membranes prior to surface staining.

Staining was conducted using a two-step optimized protocol described by Vernay and Cosson [50]. Initially, cells were fixed in 3.7% paraformaldehyde for 10 min at 37 °C. Following fixation, cells were placed on ice and incubated with 1% bovine serum albumin (BSA) for 45 min to block nonspecific antibody binding. For surface marker analysis, no permeabilization was performed. To assess whole-cell protein expression, a second fixation was carried out on ice, followed by permeabilization with 0.05% Triton X-100 (Merck) for 10 min before applying primary and secondary antibodies as described by Vernay and Cosson [50].

Three different unconjugated anti-ChAT primary antibodies were used for confocal imaging: a monoclonal mouse anti-ChAT antibody (MAB3447, R&D Systems, Bio-Techne, Stockholm, Sweden), which targets a region between Ala2–Pro630 (Accession # NP_066266); a rabbit polyclonal anti-ChAT antibody (PAB14536, Abnova, Taipei, Taiwan); and another rabbit polyclonal anti-ChAT antibody (AB143, Millipore via Merck). Additional primary antibodies included mouse monoclonal anti-AChE (A-11, Santa Cruz), mouse monoclonal anti-BChE (D-5, Santa Cruz Biotechnology (Dallas, TX, USA)), rat monoclonal anti-α7 nicotinic AChR (Santa Cruz Biotechnology), and mouse monoclonal anti-M1 muscarinic AChR (Santa Cruz Biotechnology). All primary antibodies were unconjugated and used at optimized dilutions ranging from 1:50 to 1:300.

Secondary antibodies were fluorophore-conjugated and included goat anti-mouse IgG (FITC, Invitrogen, 1:375), goat anti-rabbit IgG (Alexa Fluor 647, Invitrogen, 1:300), and goat anti-rat IgG (Alexa Fluor 647, Invitrogen, 1:300). Nuclear staining was performed using NucBlue DAPI reagent (Invitrogen), and mitochondrial labeling was achieved with Mitotracker Red CMXRos (Invitrogen).

Confocal imaging was carried out using a Nikon Ti2 inverted microscope equipped with multipoint spinning disk and DeepSIM super-resolution capabilities. Z-stack images were captured using 405, 477, 545, and 637 nm lasers. To ensure consistency across experiments, microscope settings—including exposure time—were kept constant for all samples, whether stained with primary and secondary antibodies (Figure 7 and Figure 8, Appendix A) or with secondary antibodies alone (Appendix A).

## 5. Conclusions

In conclusion, our findings—consistent with growing evidence—indicate that a functional cholinergic phenotype is a common feature among several carcinoma cell lines, suggesting potential avenues for therapeutic exploitation. Notably, the identification of an extracellular, membrane-bound form of ChAT uniquely in neuroblastoma SH-SY5Y cells reveals a previously unrecognized mode of in situ ACh signaling. This novel observation warrants further investigation to elucidate its biological significance and potential implications in cancer cell communication and survival.

## Figures and Tables

**Figure 1 ijms-26-10311-f001:**
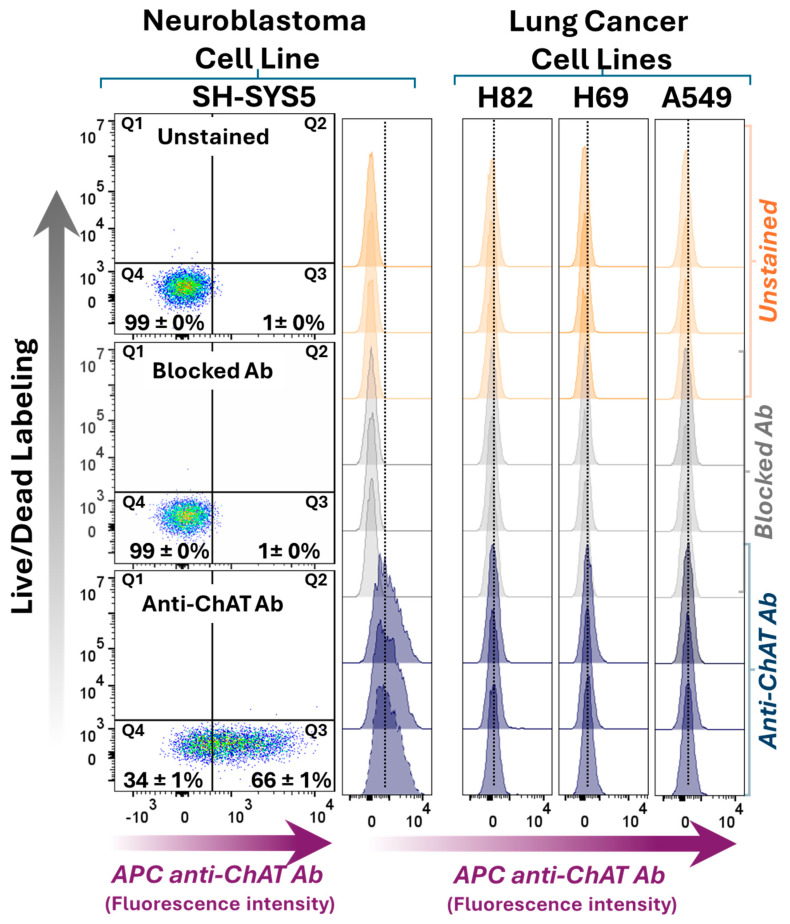
Surface ChAT staining and flow cytometric analyses in a neuroblastoma cell line compared to three lung cancer cell lines. The left panel shows dot plots of Live/Dead dye staining versus ChAT staining in SH-SY5Y neuroblastoma cells. The upper dot plot shows unstained cells, the middle plot shows results after blocking the antibody with recombinant human ChAT protein, and the lower plot indicates the percentage of SH-SY5Y cells positive for ChAT staining using the ChAT antibody alone. The values in the dot plots represent mean ± SD of triplicates. The right panel presents histograms for the small cell lung cancer lines H82 and H69, and the lung adenocarcinoma cell line A549, stained similarly to the left panel. Notably, plasma membrane integrity was also assessed in the lung cancer cell lines using Live/Dead dye staining. All cells exhibited intact plasma membranes, as no major signal was detected in quadrants Q1 and Q2. Abbreviations: ChAT = choline acetyltransferase; APC anti-ChAT Ab = allophycocyanin-conjugated anti-ChAT antibody.

**Figure 2 ijms-26-10311-f002:**
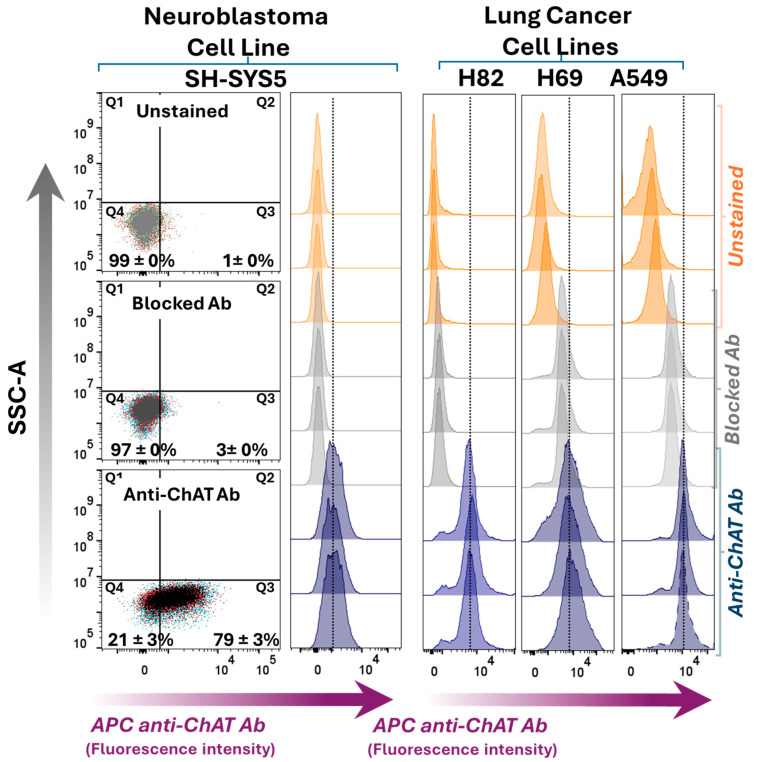
Whole-cell ChAT staining and flow cytometric analyses in a neuroblastoma cell line compared to three lung cancer cell lines. The left panel shows dot plots of side scatter area versus ChAT staining in SH-SY5Y neuroblastoma cells following membrane permeabilization, allowing the anti-ChAT antibody to access both surface and intracellular ChAT protein, thereby reflecting whole-cell ChAT staining. The upper dot plot shows unstained cells, the middle plot shows cells incubated with a pre-blocked anti-ChAT antibody, and the lower plot indicates the percentage of SH-SY5Y cells positive for ChAT expression. The right panel presents histograms for the small cell lung carcinoma lines H82 and H69, and the lung adenocarcinoma cell line A549, analyzed similarly to the left panel. Notably, ChAT staining in the lung carcinoma cell lines reflects only intracellular localization, as these cells lacked surface-localized ChAT (as shown in Figure 1). A comparison of ChAT staining in SH-SY5Y cells (79 ± 3%) with the extracellular ChAT data in Figure 1 (66 ± 1%) indicates that, in contrast to the lung carcinoma cell lines, most of the ChAT protein in SH-SY5Y cells is localized extracellularly. The values in the dot plots represent mean ± SD of triplicates. ChAT = choline acetyltransferase; APC anti-ChAT Ab = allophycocyanin-conjugated anti-ChAT antibody.

**Figure 3 ijms-26-10311-f003:**
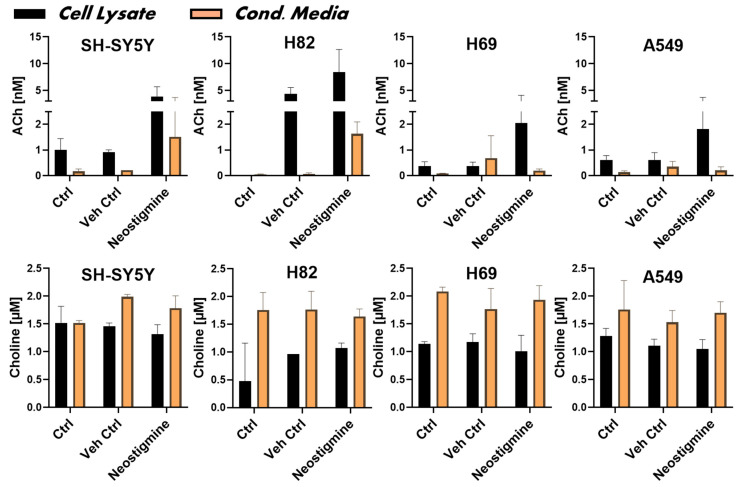
Synthesis and release of acetylcholine (ACh) by the neuroblastoma cell line and three small cell lung cancer cell lines. The upper panel displays the measured concentrations of ACh in both cell lysates and conditioned media, representing intracellular biosynthesis and extracellular release, respectively. Control cells (Ctrl) refer to untreated samples, while vehicle controls (Veh Ctrl) were treated with dimethyl sulfoxide (DMSO). Neostigmine, a dual inhibitor of the ACh-hydrolyzing enzymes acetylcholinesterase (AChE) and butyrylcholinesterase (BChE), was used to prevent enzymatic degradation of ACh, enabling accurate quantification via HPLC-MS/MS. Conditioned media were collected as described in Section 4.4 of the Materials and Methods. Data are presented as mean ± standard deviation (SD). Abbreviations: AChE = acetylcholinesterase; BChE = butyrylcholinesterase; DMSO = dimethyl sulfoxide.

**Figure 4 ijms-26-10311-f004:**
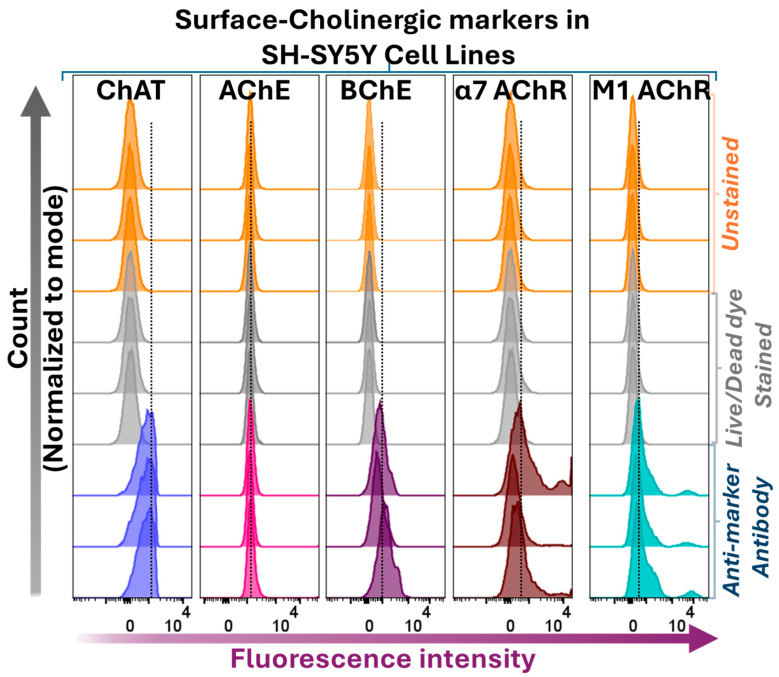
Surface staining and flow cytometric analyses of ChAT in relation to other cholinergic markers in the SH-SY5Y neuroblastoma cell line. Histograms show surface staining for AChE, BChE, nicotinic α7 AChR, and muscarinic M1 AChR in relation to surface ChAT staining in SH-SY5Y cells. Notably, the cells express all assessed components of the cholinergic machinery at their surface, except for AChE. Histograms compare shifts in fluorescence intensities of cells stained with marker-specific antibodies relative to unstained controls and Live/Dead dye-stained cells. Live/Dead dye staining was performed to confirm that the plasma membrane remained intact, ensuring that antibody signals originated exclusively from surface-localized targets. Abbreviations: ChAT = choline acetyltransferase; AChE = acetylcholinesterase; BChE = butyrylcholinesterase; α7 AChR = α7-subtype of nicotinic acetylcholine receptor; M1 AChR = M1-subtype of muscarinic acetylcholine receptor.

**Figure 5 ijms-26-10311-f005:**
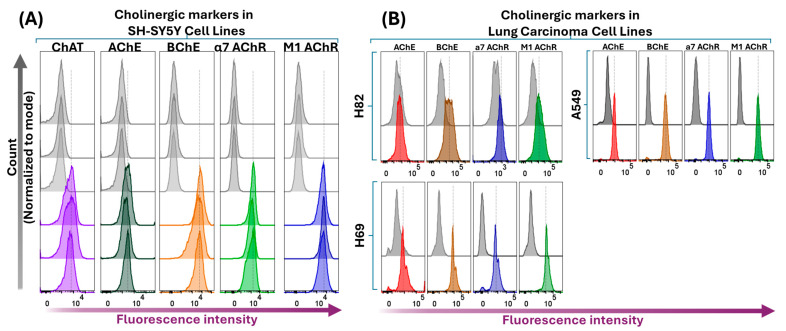
Whole-cell-stained flow cytometric analyses of various cholinergic signaling components in the SH-SY5Y neuroblastoma cell line compared to lung carcinoma cell lines. Panel (**A**) shows histogram data depicting total intracellular and surface protein expression in SH-SY5Y cells stained after membrane permeabilization, reflecting whole-cell staining of ChAT, AChE, BChE, α7 AChR, and M1 AChR, respectively. Grey histograms represent unstained cells, while colored histograms show staining of the individual cholinergic components. Notably, whole-cell staining revealed that, upon membrane permeabilization, the anti-AChE antibody positively stained the cells, suggesting that AChE is either primarily located intracellularly or exists as an isoform that is secreted as a free extracellular enzyme. Panel (**B**) presents corresponding whole-cell staining data for lung carcinoma cell lines H82, H69, and A549, with data presented as in panel (**A**). The results clearly show that these cell lines also express all tested cholinergic receptors and enzymes, including AChE. ChAT = choline acetyltransferase; AChE = acetylcholinesterase; BChE = butyrylcholinesterase; α7 AChR = α7-subtype of nicotinic acetylcholine receptor; M1 AChR = M1-subtype of muscarinic acetylcholine receptor.

**Figure 6 ijms-26-10311-f006:**
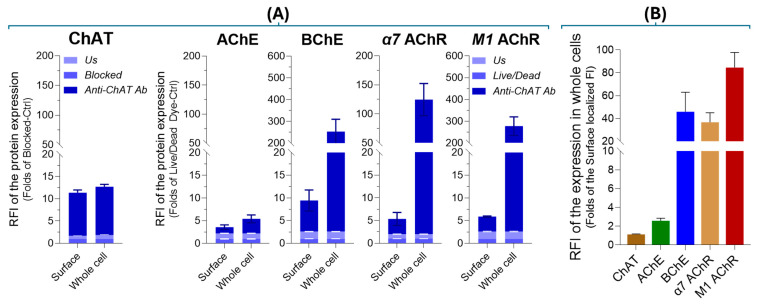
Relative surface and total protein expression of cholinergic markers in SH-SY5Y neuroblastoma cells. Protein levels were quantified based on fluorescence intensity (FI) obtained from surface and whole-cell flow cytometry data (Figure 4 and Figure 5). Panel (**A**) presents relative fluorescence intensity (RFI) values normalized to controls: for ChAT, normalization was performed against cells incubated with a blocked anti-ChAT antibody, while for AChE, BChE, α7-nAChR, and M1 mAChR, normalization was done using cells stained only with the Live/Dead dye. Panel (**B**) depicts the fold difference between total cellular and surface-localized protein expression, calculated by dividing FI from whole-cell staining by FI from surface staining. Bar graphs represent the mean ± standard deviation (SD). Abbreviations: ChAT = choline acetyltransferase; AChE = acetylcholinesterase; BChE = butyrylcholinesterase; α7 AChR = α7-subtype of nicotinic acetylcholine receptor; M1 AChR = M1-subtype of muscarinic acetylcholine receptor.

**Figure 7 ijms-26-10311-f007:**
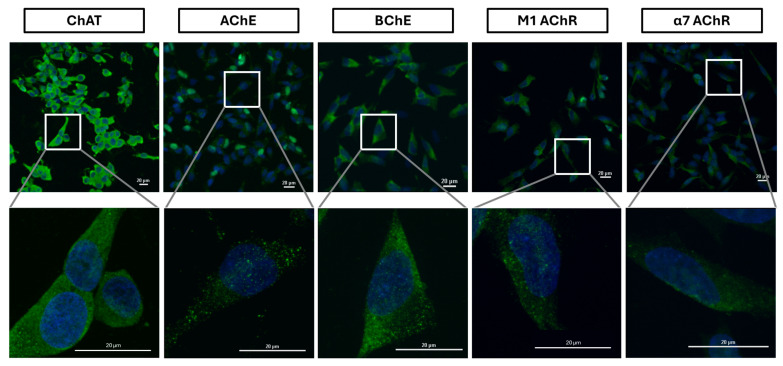
Surface confocal microscopy of cholinergic markers in SH-SY5Y neuroblastoma cells. Cells were incubated with specific antibodies under non-permeabilizing conditions to selectively stain extracellularly localized proteins. Representative micrographs show surface immunoreactivity for ChAT, AChE, BChE, α7-nAChR, and M1-mAChR (green), co-labeled with DAPI to visualize nuclei (blue). Images were acquired at 20× magnification (scale bar: 20 µm), with corresponding zoomed-in views at 60× magnification highlighting regions indicated by white boxes. Abbreviations: ChAT = choline acetyltransferase; AChE = acetylcholinesterase; BChE = butyrylcholinesterase; α7-nAChR = α7-subtype of nicotinic acetylcholine receptor; M1-mAChR = M1-subtype of muscarinic acetylcholine receptor.

**Figure 8 ijms-26-10311-f008:**
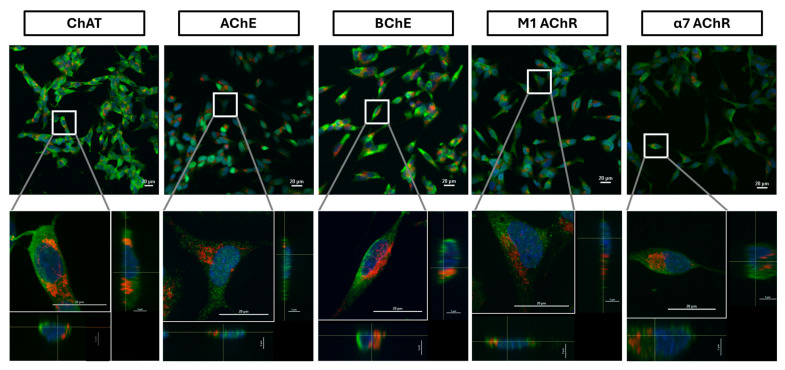
Extracellular and intracellular staining of cholinergic markers in the SH-SY5Y neuroblastoma cell line. Cells were subjected to membrane permeabilization followed by incubation with specific antibodies, allowing detection of both surface and intracellular protein expression. Micrographs show total staining signals for cholinergic markers (ChAT, AChE, BChE, M1 AChR, and α7 AChR) in green, co-labeled with DAPI for nuclei (blue) and Mitotracker Red for mitochondria (red). Images were acquired at 20× magnification (scale bar: 20 µm), with corresponding zoomed-in views at 60× magnification from regions indicated by white boxes. Side views of X-Y cross sections from the Z-stack are displayed at the bottom and right of each panel, illustrating intra- and extracellular localization of the antibody-associated fluorescent signals. Abbreviations: ChAT = choline acetyltransferase; AChE = acetylcholinesterase; BChE = butyrylcholinesterase; α7-nAChR = α7-subtype of nicotinic acetylcholine receptor; M1 AChR = M1-subtype of muscarinic acetylcholine receptor.

**Figure 9 ijms-26-10311-f009:**
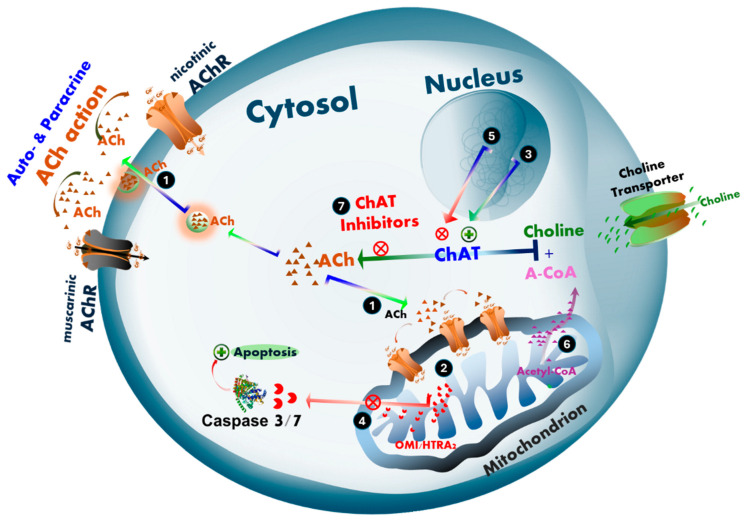
Hypothetical mechanisms of intracellular cholinergic signaling in the regulation of cell survival and proliferation in cancer, and neurodegeneration in Alzheimer’s disease (AD) and amyotrophic lateral sclerosis (ALS). (1) Acetylcholine (ACh) is proposed to promote cell proliferation and/or survival through an intracellular signaling mechanism that involves monitoring mitochondrial bioenergetic function. (2) This mechanism includes the regulation of mitochondria-driven apoptosis [48], which may be differentially affected during the progression of cancer and neurodegenerative diseases such as AD and ALS. (3) In cancer, tumor cells may upregulate the expression and activity of choline acetyltransferase (ChAT), leading to elevated intracellular ACh levels. (4) This upregulation may serve as a protective mechanism against cellular damage induced by treatments such as radiotherapy or chemotherapy, by suppressing mitochondria-driven apoptosis [20,21,22,23,24,25,26,41]. This suppression may allow more time for DNA repair, thereby enhancing cell survival, promoting metastatic spread, recolonization of the tumor site, and potentially contributing to cancer relapse. (5) In contrast, in cholinergic neurodegenerative disorders such as AD and ALS, a progressive decline in ChAT expression is observed in neurons [6,7,8,9,10,15,16]. This decline may result from aging, reduced stimulation or availability of neurotrophic factors (e.g., NGF, BDNF), or the inappropriate use of anticholinergic drugs, including proton pump inhibitors (PPIs) and various receptor antagonists. (6) Additionally, metabolic and mitochondrial dysfunction may reduce the production or cytoplasmic availability of acetyl-Coenzyme A (A-CoA), a critical co-factor required in equimolar amounts for ACh synthesis by ChAT. A-CoA is primarily generated in mitochondria, where it is used for ATP production via the Krebs cycle and exported to the cytoplasm for biosynthetic processes, including ACh synthesis. A reduction in A-CoA availability may impair intracellular ACh biosynthesis, weakening the cell’s defense against mitochondria-driven apoptosis and rendering cholinergic neurons more vulnerable to degeneration and anticholinergic insults. (7) This dual-edged mechanistic hypothesis can be experimentally tested using selective ChAT inhibitors, such as PPIs [35], and warrants further investigation. The ACh–mitochondria–apoptosis axis may represent a critical pathway for understanding both selective cholinergic neurodegeneration in dementia and neuromotor disorders, and the survival strategies exploited by cancer cells.

## Data Availability

The datasets presented in this article are not readily available because the data are part of an ongoing study. Requests to access the datasets should be directed to the corresponding author.

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
