# Peer review of "Comparative Analysis of Cholinergic Machinery in Carcinomas: Discovery of Membrane-Tethered ChAT as Evidence for Surface-Based ACh Synthesis in Neuroblastoma Cells"

_ijms, 2025, doi:10.3390/ijms262110311_

Round 1

Reviewer 1 Report

Comments and Suggestions for Authors

The MS presents interesting findings from a study of what could be termed “neurochemical correlates of cholinergic neurotransmission” in neuroblastoma (and other tumour) cells. Some of the results could have potential implications for how we understand the physiology of cancer. The approach is fundamentally sound; the scope of the outcomes could be limited by the fact that the study is based on cultured cells only rather than on an experimental model closer to “real life”. However, under the circumstances (as an initial observation) it can probably pass all right. I do have comments and queries, though, as specified below:

#   1   As somebody who spent a lot of time working with neuroblastoma cells, I am aware of the limitations some of their lines have, e.g. inconstant karyotype, variable/capricious expression of neurotransmitters depending on culturing conditions etc. and I tend to look at any study using neuroblastoma cells with a degree of scepticism if not outright suspicion. SH-SY5Y is a human cell line that may not have the problems mentioned above. I wonder whether the choice of SH-SY5Y should not be pointed out and discussed in a little more detail (Materials and Methods?) and the reasons for the choice spelled out, perhaps also in Abstract.

#   2   Line 26 (and elsewhere, see also # 7)): We are looking for experimental data and not “expecting” anything – unless, perhaps, when something is specifically outlined in a hypothesis and looking very probable indeed, but even then, we should not be “surprised” (cf. # 11) and just accept the result.

#   3   Line 53: I would say “has been considered” rather than “is” Effectively, as I understand it, you are testing & challenging the “is” view.

#   4   Lines 72/73: “progressively” I wonder whether I would not add “selectively” to be quite precise: neither the γ-motor neurons, nor the cholinergic innervation of the extraocular muscles are lost, paralysis of the intercostal muscles comes often quite late, if at all.

#   5   Line 138/139: “a major proportion of ChAT protein.. “ Not necessarily true, since authors have not quantitatively estimated the actual amounts of ChAT protein. Figs 1 and 2 show numbers of cells, forward scatter v. side scatter (am I guessing correctly?). One would have to somehow estimate the actual amounts of ChAT protein in the cytosol versus that on the cell surface (see also # 10).

#   6   Line 168 and elsewhere (incl. Fig 3) “conditioned medium”. It should be specified in Materials and Methods (and possibly mentioned in Legend of Fig 3) how the “conditioned” medium was obtained.

#   7   Lines 188, 191, 223, 228, 229 and elsewhere: gratuitous insertions of “as expected”, “expected” etc. could make readers feel suspicious; at the very least, the repeated use of these expressions is incongruous with the empirical nature of the study. If you are testing hypotheses in an impartial (non-biased) manner, then you do not “expect” anything, you are just keen to see the result. Most of these “expectations” could be dropped as superfluous.

#   8   Line 223: “partially unexpected” or rather “partially expected”? This further illustrates the problem under # 7. I would prefer to see clearly formulated hypotheses, followed by tests & results, and the conclusion(s) presented, preferably in the context of broader (previous or “canonical”) knowledge, in Discussion.

#   9   Line 244 “confirm” suggest finality of the finding; “corroborate” would be more consistent with “adding to” or “strengthening of” the evidence and would be more suitable in the context (see also the next comment).

# 10   Figs 6 & 7: “Surface confocal microscopy” I am not sure that this evidence is very convincing and either “confirms” or “corroborates” the flow cytometry data. The labels in bottom panels appear to spread at some distance form the plasma membrane, even in Fig 7 (does this represent deconvoluted images?). In Fig 7 one would like to clearly see horizontal optical sections (mid-sections, z-stacks) and be convinced that the relevant label is indeed close to, or exclusively present at, the plasma membrane. Even if it were, the resolution would not allow readers to conclude at which side of the membrane the antigen resides (the thickness of the plasma membrane is probably well below the resolution capability of confocal microscopy). This is of fundamental importance, in need of corroborating studies such as marking the outer surface proteins in whole cells (perhaps biotinylation using a form of biotin which does not cross the membrane and does not enter the cells?) followed by homogenization of the cells and separate estimations of the cytosolic (unmarked) and membrane-bound (marked) protein.

#  11  Line 363: I would drop the emotionally loaded “surprising” and re-arrange the statement as “This finding, if confirmed by additional experiments, e.g. using selective marking of cell-surface bound proteins, would be very significant, particularly because the current view..”

#  12  Line 366 Can it be specified which blood cells? Perhaps,  one may consider adding, if available, references more recent than [39] which is from 1999 and might not be quite up to date.

Author Response

Please see the attached Response-to-Reviewers file.

Reviewer 2 Report

Comments and Suggestions for Authors

This manuscript provides data supporting the presence of ChAT in the cell membrane of a neuroblastoma cell line, SH-SY5Y, which could potentially benefit future studies. However, some additional clarification would be really appreciated prior to publication. See more details below.

I have to say that I am not sure some of the results are fully supported by the data provided. For example, lines 136-138 indicate a major PROPORTION of ChAT protein based on flow cytometry data showing NUMBERS of cells positive for different ChAT staining (emphasis in capital font). Yet, there seems to be a big jump there without demonstrating the staining intensity per cell, i.e., ChAT protein amount per cell. Similarly, lines 136-138 and 156-157 state that MOST of the ChAT protein in SH-SY5Y was localized extracellularly (emphasis in capital font). Yet, the corresponding supporting data only shows the percentage of cells expressing ChAT. There is a possibility that a lower percentage of positive cells is accompanied by each of these positive cells having a significantly higher expression level, thus balancing out the overall protein amount.

Another example is comparing the expression of two different proteins, for example, AChE and BChE. As noted by the authors in lines 382-384, while the used antibodies can specifically bind to AChE and BChE, their affinities for their target proteins can be vastly different. Also, three different secondary antibodies were used, i.e., anti-mouse, anti-rat, and anti-rabbit. Their binding affinities to the primary antibodies will affect the obtained intensity, too. Even for the anti-mouse secondary antibody used for the AChE and BChE staining, its binding to the primary antibodies of AChE and BChE may be different, too, since we do not know if the Fc fragments, where the secondary antibody binds to, are identical. These will render the comparison of the staining intensity of AChE with that of BChE or another protein moot. Some real-time PCR data may be helpful here. But I don’t think pure antibody staining data can support the current conclusions, for example, as presented in lines 242-244, 291-292, 300-301, 371-373, 377-378.

Another major concern I have is how SH-SY5Y, more particularly, the SH-SY5Y tested as described in the current manuscript, is representative of neuroblastoma. First of all, SH-SY5Y is an adherent cell line. Yet, all of the presented experiments, including the immunofluorescent staining detected by confocal microscopy, use trypsin treatment. See lines 510-511, 514, and 550. Does this trypsination step change anything? Is it possible to culture the adherent cells on coverslips and perform immunofluorescent staining on those coverslips without any trypsination? Further, cancer cell lines may not be representative of cancer cells in patients. Without human cancer tissue data, it is suggested to add some quick sentences discussing this limitation.

Concerning Figure 2, it seems the blocking step for H69 and A549 was not quite successful.

Concerning the neostigmine data as well as its conclusion in lines 191-193, is there theoretically a higher level of choline without neostigmine vs with neostigmine, since AChE/BChE breaks down acetylcholine into choline and acetate?

Concerning AChE expression, the flow cytometry data (Figure 4) do not seem consistent with the confocal data (Figure 6). Kindly elaborate on this finding.

Additionally, while the mitochondria-related/neurodegenerative-disorder-related hypothesis presented in the discussion section is interesting, I am not sure it is closely/directly related to the presented results. Accordingly, it is suggested to delete Figure 8 and shrink the paragraph starting in line 439.

While I was reading the manuscript, some super minor formality issues were noticed.

(1) Some brackets do not appear in pairs. See, lines 16, and 506-507.

(2) Citations are needed for line 130.

(3) A “.” is missing in line 257.

(4) Using of “SH-SY-5Y” instead of “SH-SY5Y” in line 150.

Author Response

(The authors gave the same response as above.)

Reviewer 3 Report

Comments and Suggestions for Authors

In this manuscript Thakur et al have performed comparitive analysis of cholinergic system in various carcinoma cell lines. The authors have compared the expression of various cholinergic molecules such as ChAT, choline, acetylcholine, AChE and BChE among others to determine the importance of cholinergic system in cancer. 

The manuscript has important weaknesses:

1) The authors detect the expression of ChAT on the surface and within the cells in neuroblastoma and lung cancer cells. However, it is imperative to identify the expression of cholinergic molecules on normal, non-cancerous cells. This is especially important to determine how the expression of these receptors change when cells transform and cancerous. This is a major limitation of this study.

2) The purpose of checking surface vs intracellular expression of cholinergic molecules is rather less interesting as the authors have not performed any experiments to target these receptors to prevent cancer progression.

3) In Figure 6, the confocal microscopy data is unconvincing. The authors claim that ChAT os expressed on the surface. However, the confocal images suggest that ChAT is expressed in the cytoplasm of the cells.

Author Response

(The authors gave the same response as above.)
